# Improving Perturbation-based Explanations by Understanding the Role of Uncertainty Calibration

**Thomas Decker**[1,2,3]     **Volker Tresp**[2,3]     **Florian Buettner**[4,5,6]
[1]Siemens AG    [2]LMU Munich    [3]Munich Center for Machine Learning (MCML)
[4]Goethe University Frankfurt    [5]German Cancer Research Center (DKFZ)
[5]German Cancer Consortium (DKTK)
thomas.decker@siemens.com, volker.tresp@lmu.de, florian.buettner@dkfz.de

## Abstract

Perturbation-based explanations are widely utilized to enhance the transparency of machine-learning models in practice. However, their reliability is often compromised by the unknown model behavior under the specific perturbations used. This paper investigates the relationship between uncertainty calibration - the alignment of model confidence with actual accuracy - and perturbation-based explanations. We show that models systematically produce unreliable probability estimates when subjected to explainability-specific perturbations and theoretically prove that this directly undermines global and local explanation quality. To address this, we introduce ReCalX, a novel approach to recalibrate models for improved explanations while preserving their original predictions. Empirical evaluations across diverse models and datasets demonstrate that ReCalX consistently reduces perturbation-specific miscalibration most effectively while enhancing explanation robustness and the identification of globally important input features.

## 1   Introduction

The ability to explain model decisions and ensure accurate confidence estimates are fundamental requirements for deploying machine learning systems responsibly [34]. Perturbation-based techniques [54] have been established as a popular way to enhance model transparency in practice [6, 14]. Such methods systematically modify input features to quantify their importance by evaluating and aggregating subsequent changes in model outputs [11]. This intuitive principle and the flexibility to explain any prediction in a model-agnostic way has led to widespread adoption across various domains [31].

Nevertheless, the application of perturbation-based techniques faces a fundamental challenge: These methods operate by generating inputs that differ substantially from the training distribution [26, 20], and models often produce invalid outputs for such perturbed samples [18, 8, 32]. Consequently, forming explanations by aggregating misleading predictions under perturbations can significantly distort the outcome and compromise its fidelity. On top of that, this can also contribute to frequently observed instabilities of perturbation-based explanations [1, 4, 38, 13], which reduces their effectiveness and could even be exploited for malicious manipulations [58, 3].

These issues naturally raise the question of how to attain reliable model outputs under the specific perturbations used when deriving explanations. A classical approach for this purpose is uncertainty calibration [46, 45, 25]. It aims to ensure that a model's confidence aligns with its actual accuracy, which is crucial to obtaining meaningful probabilistic predictions. Consider, for instance, the situation in Figure 1 where a classifier detects a bee in the image with 99% confidence. Then this value is only reliably interpretable if for all predictions with corresponding confidence, precisely 99% are indeed correct. While calibration has been extensively studied in the machine learning literature [63], its

39th Conference on Neural Information Processing Systems (NeurIPS 2025).

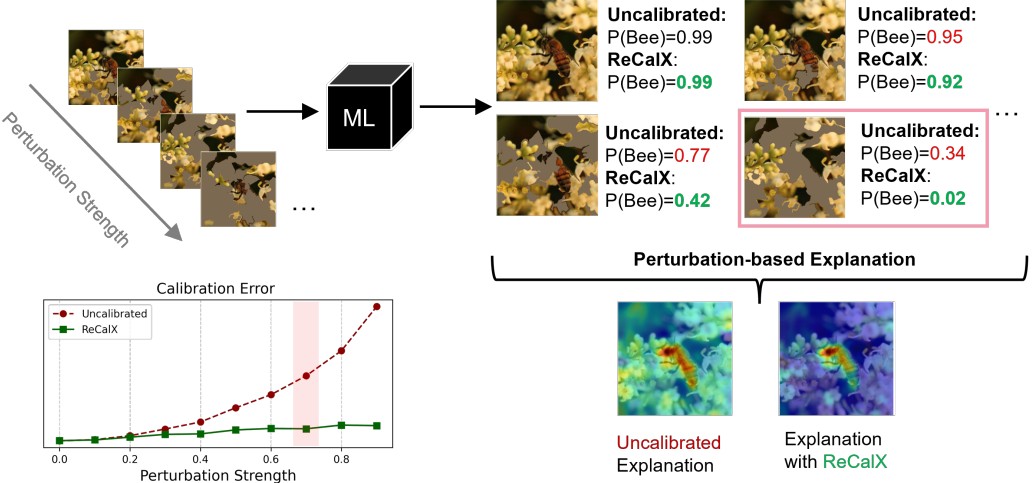

Figure 1: Perturbation-based explanation methods typically query the model on modified inputs and aggregate the resulting prediction changes to identify relevant features. However, we show that models typically produce significantly miscalibrated output probabilities under commonly used perturbations. This means that the underlying predictions used to derive explanations do not reflect actual changes in class likelihoods, obscuring true feature importance. To mitigate this, we propose ReCalX as a simple recalibration technique that enables reliable outputs under explainability-specific perturbations, leading to more informative explanation results.

role in explanation methods remains largely unexplored. First empirical evidence suggests that basic calibration might indeed benefit explainability [55, 39, 40], but a rigorous theoretical understanding is still missing. In this work, we provide the first comprehensive analysis of the relationship between uncertainty calibration and perturbation-based explanations. Our findings establish calibration as a fundamental prerequisite for reliable model explanations and propose a practical solution for enhancing the quality of perturbation-based explanation methods (see Figure 1).

More precisely, we make the following contributions:

- We provide a rigorous theoretical analysis revealing how poor calibration can bias the results of common perturbation-based explainability techniques.

- We show that common neural classifiers for tabular and image data exhibit high levels of miscalibration under explainability-related perturbations.

- We propose ReCalX, a novel approach that increases the reliability of model outputs under the particular perturbations used to derive explanations and validate its effectiveness.

- We demonstrate that after applying ReCalX, explanation results are significantly more robust and better identify input features that are relevant for high performance.

## 2 Background and Related Work

**Basic Notation**   Consider a probability space $(\Omega, \mathcal{F}, P)$, where $\Omega$ is the sample space, $\mathcal{F}$ is a $\sigma$-algebra of events, and $P$ is a probability measure. Let $X : \Omega \to \mathcal{X}$ be a random variable representing the feature space $\mathcal{X}$, and let $Y : \Omega \to \mathcal{Y}$ be a random variable representing the target space $\mathcal{Y}$. The joint data distribution of $(X, Y)$ is denoted by $P_{X,Y}$, the conditional distribution of $Y$ given $X$ by $P_{Y|X}$, and the marginal distributions by $P_X, P_Y$. During our theoretical analysis, we will also make use of the following quantities. The *mutual information* between two random variables $X$ and $Y$, measures the reduction in uncertainty of one variable given the other and is defined as:

$$I(X, Y) := \mathbb{E}_{(X,Y)} \left[ \log \frac{P_{X,Y}(X, Y)}{P_X(X) P_Y(Y)} \right].$$

A related measure is the *Kullback-Leibler (KL) divergence*, which expresses the difference between two probability distributions $P$ and $Q$ over the same sample space:

$$D_{\mathrm{KL}}(P\|Q) := \mathbb{E}_X\left[\log\frac{P(X)}{Q(X)}\right].$$

**Perturbation-based Explanations**  Perturbation-based explanations quantify the importance of individual input features by evaluating how a model output changes under specific input corruptions. Let $f : \mathcal{X} \to [0,1]^K$ be a classification model over $K$ classes, where $\mathcal{X} \subseteq \mathbb{R}^d$ is the $d$-dimensional input space. Given a sample $x \in \mathcal{X}$, a perturbation-based explanation produces an importance vector $\phi(x) \in \mathbb{R}^d$, where $\phi_i(x)$ quantifies the importance of feature $x_i$. To derive $\phi(x)$, we first introduce a perturbation function $\pi : \mathcal{X} \times 2^{\{1,\ldots,d\}} \to \mathcal{X}$ that, given an instance $x$ and a subset of feature indices $S \subseteq \{1,\ldots,d\}$, returns a perturbed instance $\pi(x,S)$ where features in $S$ remain unchanged while the complementary features are modified. Various perturbation strategies have been investigated [60, 9, 11], with common approaches being the replacement of features with fixed baseline values [32] or blurring [19, 33] for images. The model prediction under such perturbation is given by:

$$f_S^\pi(x) := f\big(\pi(x,S)\big).$$

To compute the final explanation $\phi$, the outcomes of $f_S$ for different subsets $S$ are systematically examined and aggregated. While numerous strategies exist for this purpose [11, 49, 17, 47, 66, 53, 42], we focus on methods that employ a linear summary strategy [38]. In this case, there exists a summary matrix $A \in \mathbb{R}^{d \times 2^d}$ such that $\phi(x) = A\vartheta(x)$, where $\vartheta \in \mathbb{R}^{2^d}$ contains the predictions $f_S(x)$ for all possible subset perturbations $S$. Two prominent methods in this category are Shapley Values [42] and LIME [53].

**Uncertainty Calibration**  In general, a model is said to be *calibrated* if its predicted confidence levels correspond to empirical accuracies for all confidence levels. Formally, this calibration property requires that $P_{Y|f(X)} = f(X)$ [7, 59, 24]. In classification, this intuitively implies, that for any confidence level $p \in [0,1]$ it holds:

$$P(Y = k \mid f(X)_k = p) = p \quad \forall k \in \{1,\ldots,K\},$$

where $f(X)_k$ denotes the predicted probability for class $k$ and $Y$ is the true label. This means that among all instances for which the model predicts a class with probability $p$, approximately a $p$ fraction should be correctly classified. In practice, modern neural classifiers often exhibit miscalibration [25, 44], in particular when facing out-of-distribution samples [48, 61, 65]. This can be formalized using a *calibration error (CE)*, which quantifies the mismatch between $f(X)$ and $P_{Y|f(X)}$. In this work, we will consider the KL-Divergence-based calibration error [51], as this is the proper calibration error directly induced by the cross-entropy loss which is typically optimized for in classification:

$$CE_{KL}(f) = \mathbb{E}\Big[D_{KL}\Big(P_{Y|f(X)}\|f(X)\Big)\Big]$$

Our work investigates how violations of this calibration property affect the quality of the derived explanations $\phi(x)$ and develops methods to improve calibration specifically for perturbation-based techniques, which has not been analyzed before.

## 3  Understanding the Effect of Uncertainty Calibration on Explanations

In this section, we provide a thorough mathematical analysis of how poor calibration under explainability-specific perturbation can degrade global and local explanation results. All underlying proofs and further mathematical details are provided in Appendix A.

**Effect on Global Explanations**  Calibration is inherently a global property of a model related to the entire data distribution and in general, it is impossible to evaluate it for individual samples [69, 43]. Hence, we first analyze its effect on global explanations about the model behavior [41] such as the importance of individual features for overall model performance [12, 15]. Following [12], we define the global predictive power of a feature subset $S$ for a model $f$, as follows:

**Definition 3.1.** Let $S \subset \{1, \ldots, d\}$ be a feature subset and let $f_S^\pi$ be the model that only observes features in $S$ while the other are perturbed using a predefined strategy $\pi$, then the predictive power $v_f^\pi$ (S) for a model $f$ and a loss function $\mathcal{L} : \mathcal{Y} \times \mathcal{Y} \to \mathbb{R}^+$ is defined as:

$$v_f^\pi(S) = \mathbb{E}[\mathcal{L}(f_\emptyset^\pi(X), Y)] - \mathbb{E}[\mathcal{L}(f_S^\pi(X), Y)]$$

Note, that the predictive power captures the performance increase resulting from observing features in $S$ compared to a baseline prediction where all features are perturbed using $\pi$. In classification, models are typically optimized for the cross-entropy loss $\mathcal{L}_{CE}(f(x), y) = -\log(f(x)_y)$. We show that in this case the predictive power can be decomposed as follows:

**Theorem 3.2.** *Let $\mathcal{L}_{CE}$ be the cross-entropy loss, $D_{KL}(\cdot, \cdot)$ be the KL-Divergence between two distributions, and let $I(\cdot, \cdot)$ denote the mutual information between random variables. Then we have:*

$$v_f^\pi(S) = \underbrace{D_{KL}(P_Y \| f_\emptyset^\pi(X))}_{\text{Perturbation Baseline Bias}} + \underbrace{I(f_S^\pi(X), Y)}_{\text{Information in } f_S^\pi \text{ about } Y} - \underbrace{CE_{KL}(f_S^\pi)}_{\text{Calibration Error of } f_S^\pi}$$

This theorem is in line with existing decomposition results [7, 36, 24] that hold for a general class of performance measures related to proper scoring rules [21], including $\mathcal{L}_{CE}$. It further allows for an intuitive interpretation of what drives the predictive power when evaluated with a specific perturbation and reveals a fundamental relationship with calibration. The first term can be interpreted as bias due to the perturbation strategy, implying that an optimal uninformative baseline prediction should yield $P_Y$. The second term is the mutual information between the target $Y$ and the model that only observes features in $S$. This corresponds to the reduction in uncertainty about the target $Y$ that knowledge about $f_S^\pi$ provides. The third term is the KL-Divergence-based calibration error of the restricted model $f_S^\pi$ when facing the perturbation $\pi$. The theorem implies that if the model to be explained produces unreliable predictions under the perturbation used, then the resulting calibration error directly undermines the predictive power. Moreover, Theorem 3.2 has the following immediate consequence:

**Corollary 3.3.** *If a model $f$ is perfectly calibrated under all subset perturbations faced during the explanation process, then we have:*

$$v_f^\pi(S) = I(f_S^\pi(X), Y)$$

This implies that $I(f_S^\pi(X), Y)$ can be considered as an idealized predictive power that can only be attained when the model always gives perfectly calibrated prediction under every perturbation faced. Note this also directly extends a corresponding result in [12] showing that this relationship holds for the naive Bayes classifier $P_{Y|X}$. It rather holds for any model that is perfectly calibrated under the perturbations faced, which is also the case for $P_{Y|X}$. Moreover, explaining by directly approximating similar mutual information objectives has also been proposed as a distinct way to enhance model transparency [10, 56].

**Effect on Local Explanations**   Local explanations enable the understanding of which features are important for an individual prediction. As emphasized above, calibration is a group-level property so evaluating it for a single sample is impossible. Nevertheless, we can still estimate how an overall calibration error under the perturbations might propagate to an individual explanation result:

**Theorem 3.4.** *Let $\phi(x)$ be a local explanation for input $x$ with respect to model prediction $f(x)$ and perturbation $\pi$. Let $\phi^*(x)$ denote the explanation that would be obtained if the model $f$ were perfectly calibrated under all subset perturbations. Define the maximum calibration error across all perturbation subsets as: $CE_{KL}^{\max_S} := \max_{S \subset [d]} CE_{KL}(f_S^\pi)$ Then, with probability at least $(1 - \delta)$, the mean squared difference between the actual and ideal explanations is bounded by:*

$$\frac{1}{d} \|\phi(x) - \phi^*(x)\|_2^2 \le 2CE_{KL}^{\max_S} + \sqrt{8 \log(1/\delta)}$$

The theorem above shows that poor calibration under perturbations can at worst also directly distort the outcomes of a local explanation result $\phi(x)$ [1]. Importantly, it demonstrates that in order to improve

---

[1]Note that the particular bound presented in Theorem 3.4 holds specifically for perturbation-based methods with a bounded linear summary strategy which is satisfies by many popular choices such as Shapley Values and LIME. In Appendix A we provide a more general version of this theorem that equivalently holds for non-linear aggregation approaches.

explanation quality through uncertainty calibration, it is insufficient to only reduce the calibration error on unperturbed samples, as proposed by [55]. Instead, reliable explanations explicitly require models to be well-calibrated under all subset perturbations $\pi(\cdot, S)$ used during the explanation process. In the next section, we propose an actionable technique, called ReCalX, to achieve this goal.

## 4 Improving Explanations with Perturbation-specific Recalibration

Recalibration techniques aim to improve the reliability of the probability estimates for an already trained model in a post-hoc manner. While a variety of different algorithmic approaches exist for this purpose [57], none of them is explicitly designed to enhance explanations. In this section, we introduce ReCalX as a novel and practical approach to recalibrate models to achieve both more reliable confidences and better perturbation-based explanations simultaneously.

**Information-preserving Calibration**    Recalibration methods typically transform the model's original outputs to better align predicted probabilities with empirical frequencies [57]. However, arbitrarily postprocessing predictions might distort the underlying model behavior, which we originally sought to explain. Hence, we first introduce the concept of information-preserving recalibration:

**Definition 4.1.** Let $f : \mathcal{X} \to [0,1]^K$ be a classifier and $\mathcal{T} : [0,1]^K \times 2^{\{1,\dots,d\}} \to [0,1]^K$ be a recalibration function that post-processes raw probabilities depending on the unperturbed feature subset $S$. Then $\mathcal{T}$ is information-preserving if for all feature subsets $S \subseteq \{1,\dots,d\}$:

$$I(\mathcal{T}(f_S^\pi(X), S), Y) = I(f_S^\pi(X), Y)$$

Remember Corollary 3.3. implies that $I(f_S^\pi(X), Y)$ can be considered as idealized predictive power of a subset $S$ under perfect calibration. Thus, we want any calibration method to preserve this quantity in order to attain explanations that are consistent with the original model we intended to explain. This can, for instance, be achieved using calibration maps $\mathcal{T}$ of the following kind:

**Proposition 4.2.** *Let $\mathcal{T} : [0,1]^K \times 2^{\{1,\dots,d\}} \to [0,1]^K$ be a deterministic and componentwise strictly monotonic function for every fixed $S$. Then $\mathcal{T}$ is information-preserving.*

The proof of this proposition is provided in Appendix A along with further mathematical details. Note that [68] has shown that such calibrators are also accuracy-preserving, meaning that they will not affect the original model predictions as the ranking of outcomes remains unchanged. Moreover, many existing techniques for post-hoc calibration satisfy this property [25, 37, 68, 52] and are therefore also suitable to enhance perturbation-based explanation.

**Recalibration for Explanations (ReCalX)**    Based on our theoretical analysis, an ideal recalibration method for explanations should satisfy two key requirements: (1) it should be information-preserving as defined in Definition 4.1, and (2) it should effectively reduce calibration error under all perturbations faced during the explanation process. For the first requirement, we focus on calibration methods that are componentwise strictly monotonic, and among these, temperature scaling has emerged as the most widely adopted method due to its simplicity and empirical effectiveness [25, 44]. Given a model $f : \mathcal{X} \to [0,1]^K$ that outputs logits $z(x) \in \mathbb{R}^K$ before the final softmax layer, temperature scaling introduces a single scalar parameter $T > 0$ that additionally rescales the logits:

$$f(x; T)_k := \frac{\exp(z_k(x)/T)}{\sum_{j=1}^{K} \exp(z_j(x)/T)}$$

The temperature parameter $T$ is optimized on a held-out validation set, typically by minimizing the cross-entropy loss $\mathcal{L}_{CE}$ in the case of classification. To address the second requirement, we propose to go beyond classical temperature scaling and introduce ReCalX as a generalized version. It aims to reduce the calibration error under all perturbations faced during the explanation process by scaling logits using an adaptive temperature that depends on the perturbation level implied by $S$. Formally, for a subset $S \subseteq \{1,\dots,d\}$, we define the perturbation level $\lambda(S)$ as the fraction of perturbed features: $\lambda(S) = (d - |S|)/d \in [0,1]$. To account for different perturbation intensities, we partition $[0,1]$ into $B$ equal-width bins and learn a specific temperature for each bin. Given a validation set $\mathcal{D}_{\text{val}} = (x_i, y_i)_{i=1}^{N}$, we optimize a temperature $T_b$ for each bin by minimizing the cross-entropy loss $\mathcal{L}_{CE}$ on perturbed samples with corresponding perturbation levels. Note that in this way, we directly

optimize the KL-Divergence-based calibration error, while the discriminative power and ranking of predictions remain unaffected. This results in a set of temperature parameters that adapt to different perturbation intensities, enabling better calibration across all explanation-relevant perturbations. A more detailed description of the algorithm is presented in Appendix B.

During the explanation process, ReCalX infers the perturbation-level and applies the corresponding temperature $T(S)$ when making predictions:

$$f_{\text{ReCalX}}^{\pi}(x, S; \{T_b\}_{b=1}^B)_k = \frac{\exp(z_k(\pi(x,S))/T(S))}{\sum_{j=1}^K \exp(z_j(\pi(x,S))/T(S))} \qquad \forall k = 1, \ldots, K$$

where $T(S)$ selects the temperature from $\{T_b\}_{b=1}^B$ based on the bin index $b$ containing the perturbation level of $S$. This adaptive temperature scaling ensures appropriate confidence calibration for different perturbation intensities, ultimately aiming towards more reliable explanations by reducing the maximum calibration error across all perturbation subsets $CE_{KL}^{\max_S}$.

**Applying ReCalX beyond Classification**   The algorithmic implementation of ReCalX presented above relies on temperature scaling, which is specific to classification problems. However, both our theoretical framework and the underlying principle of perturbation-specific recalibration extend naturally to other predictive modeling tasks as well. For a regression setup, consider a model $f : \mathcal{X} \to P(Y)$ that outputs a probability distribution over a continuous target space $\mathcal{Y} \subseteq \mathbb{R}$. Note that all theoretical results developed in Section 3 hold equivalently when explaining probabilistic regression outputs with perturbation-based methods. Also the notion of information-preserving calibration (Definition 4.1) and the requirement for strictly monotonic transformations to preserve the model's learned information content extend directly to this setting. While temperature scaling can be used for classification, alternative techniques such as affine transformations [50] or injective variants of isotonic regression [68] satisfy the monotonicity property more generally. Therefore such techniques are also applicable to perform appropriate calibration for regression models. In Appendix C, we conducted corresponding experiments on two tabular regression datasets, which demonstrate that ReCalX effectively reduces the quantile-based calibration error [35] under explainability-specific perturbations for regression tasks as well.

## 5   Experiments

We conducted a comprehensive empirical evaluation of ReCalX considering neural classifiers on different tabular datasets [23, 22] and various computer vision models on the ImageNet `ILSVRC2012` dataset. Detailed descriptions of all experiments and further computational details are provided in Appendix B, while additional results supporting each finding are presented in Appendix C. Accompanying source code is available at `https://github.com/thomdeck/recalx`.

**Analyzing Calibration under explainability-specific Perturbations**   We first investigate how the calibration properties of neural classifiers are affected by common perturbations used in explanation methods. To implement ReCalX, we selected 200 random validation samples from each considered dataset and used 10 perturbed instances per considered perturbation level. This results in a calibration set of 2000 samples per bin. We explicitly evaluated the KL-Divergence-based calibration error using the consistent and asymptotically unbiased estimator proposed by [51] to be fully aligned with our theoretical analysis above. Each error is derived based on at least 5000 unseen samples from each dataset. Tables 1 and 2 present corresponding calibration errors for tabular and image models under different perturbation strategies. For tabular data (Table 1), we evaluate a standard Multi-Layer-Perception (MLP) as well as a tabular ResNet model [22] across four datasets using mean value replacement as perturbation. For image data (Table 2), we assess diverse architectures represented by a ResNet50 [27], a DenseNet121 [30], a ViT [16] and SigLIP [67] zero-shot model using both zero baseline and blur as perturbations. The results reveal substantial miscalibration across all models when subjected to explanation-specific perturbations. For tabular data, uncalibrated models exhibit maximum KL-divergence calibration errors ($CE_{KL}^{MAX_S}$) ranging from 0.012 to 0.863, while image models show errors between 0.037 and 0.418. Standard temperature scaling, which only calibrates using a static temperature on unperturbed samples, often fails to address this issue and can even worsen calibration under perturbations (e.g., increasing the maximum error from 0.262 to 0.306 for ViT). On the other hand our adaptive ReCalX approach consistently improves calibration with

substantial reductions in the average and maximum encountered error. In Appendix C, we present complementary results on other datasets confirming the efficacy of ReCalX.

| Dataset | Uncalibrated | | Temperature Scaling | | ReCalX | | |
|---|---|---|---|---|---|---|---|
| | $CE_{KL}^{AVG_S}$ | $CE_{KL}^{MAX_S}$ | $CE_{KL}^{AVG_S}$ | $CE_{KL}^{MAX_S}$ | $CE_{KL}^{AVG_S}$ | $CE_{KL}^{MAX_S}$ | $\downarrow_{MAX}$ (%) |
| **MLP Model** | | | | | | | |
| Electricity | 0.0423 | 0.1534 | 0.0452 | 0.1664 | **0.0097** | **0.0163** | 89.4% |
| Covertype | 0.0412 | 0.0797 | 0.0564 | 0.1115 | **0.0047** | **0.0061** | 92.3% |
| Credit | 0.2127 | 0.4763 | 0.2638 | 0.5961 | **0.0355** | **0.0533** | 88.8% |
| Pol | 0.1740 | 0.6735 | 0.1689 | 0.6521 | **0.0570** | **0.1679** | 75.1% |
| **ResNet Model** | | | | | | | |
| Electricity | 0.0032 | 0.0118 | 0.0090 | 0.0355 | **0.0017** | **0.0049** | 58.5% |
| Covertype | 0.0439 | 0.0963 | 0.0614 | 0.1413 | **0.0058** | **0.0080** | 91.7% |
| Credit | 0.0721 | 0.0830 | 0.0778 | 0.0847 | **0.0366** | **0.0773** | 6.9% |
| Pol | 0.2853 | 0.8633 | 0.3187 | 1.0173 | **0.0727** | **0.0910** | 89.5% |

Table 1: Calibration error comparison across datasets for MLP and ResNet models using mean value replacement perturbation. $CE_{KL}^{AVG_S}$ and $CE_{KL}^{MAX_S}$ represent average and maximum KL divergence-based calibration errors, across perturbation levels. Lower values indicate better calibration. The improvement column ($\downarrow_{MAX}$) shows the relative reduction in maximum calibration error achieved by ReCalX compared to uncalibrated models. ReCalX consistently outperforms both uncalibrated models and Temperature Scaling across all settings, with improvements ranging from 6.9% to 92.3%.

| Model | Uncalibrated | | Temperature Scaling | | ReCalX | | |
|---|---|---|---|---|---|---|---|
| | $CE_{KL}^{AVG_S}$ | $CE_{KL}^{MAX_S}$ | $CE_{KL}^{AVG_S}$ | $CE_{KL}^{MAX_S}$ | $CE_{KL}^{AVG_S}$ | $CE_{KL}^{MAX_S}$ | $\downarrow_{MAX}$ (%) |
| **Zero Baseline Perturbation** | | | | | | | |
| ResNet50 | 0.2240 | 0.4177 | 0.0595 | 0.1810 | **0.0092** | **0.0128** | 96.9% |
| DenseNet121 | 0.1626 | 0.3769 | 0.0965 | 0.2640 | **0.0060** | **0.0098** | 97.4% |
| ViT | 0.0936 | 0.2618 | 0.1172 | 0.3057 | **0.0045** | **0.0078** | 97.0% |
| SigLIP | 0.0721 | 0.2013 | 0.0449 | 0.1476 | **0.0130** | **0.0300** | 85.1% |
| **Blur Perturbation** | | | | | | | |
| ResNet50 | 0.2467 | 0.4158 | 0.0618 | 0.1659 | **0.0127** | **0.0139** | 96.7% |
| DenseNet121 | 0.0351 | 0.0714 | 0.0113 | 0.0259 | **0.0057** | **0.0092** | 87.1% |
| ViT | 0.0137 | 0.0365 | 0.0224 | 0.0559 | **0.0041** | **0.0072** | 80.3% |
| SigLIP | 0.0256 | 0.0547 | 0.0104 | 0.0246 | **0.0055** | **0.0098** | 82.1% |

Table 2: Calibration error comparison across models and perturbation types. $CE_{KL}^{AVG_S}$ and $CE_{KL}^{MAX_S}$ represent average and maximum KL divergence-based calibration errors, across perturbation levels. Lower values indicate better calibration. The improvement column ($\downarrow_{MAX}$) shows the relative reduction in maximum calibration error achieved by ReCalX compared to uncalibrated models. ReCalX consistently outperforms both uncalibrated scores and basic Temperature Scaling across all settings, yielding substantial improvements with relative maximum calibration error reductions of $> 80\%$.

To further validate the robustness and generality of ReCalX, we conducted several additional sensitivity and ablation experiments detailed in Appendix C. In summary, the results confirm that ReCalX remains highly effective even for advanced domain-specific perturbation methods and show that miscalibration patterns can exhibit strong correlations across different perturbation types. Furthermore, ReCalX achieves most of the potential calibration improvements with only a few hundred validation samples, and using a higher number of perturbation level bins $B$ typically yields monotonic improvements with diminishing returns beyond 10 bins.

To better understand the relationship between miscalibration and the number of perturbed features during the explanation process, we present calibration errors as a function of the perturbation level.

Figure 2 visualizes the normalized calibration error across 10 different tabular datasets. For both models, miscalibration tends to increase uniformly with higher perturbation levels. Figure 3 confirms the high variability of calibration properties across perturbation strengths, while also indicating that errors do not always grow. The ResNet50, for instance, exhibits worse miscalibration at lower perturbation levels, and we provide additional examples in Appendix C. These diverse miscalibration patterns across models and perturbation strengths highlight the need for adaptive recalibration strategies like ReCalX, which consistently reduces maximum calibration errors across all settings, significantly outperforming standard temperature scaling. Next, we want to validate whether these improvements also translate to better perturbation-based explanation results.

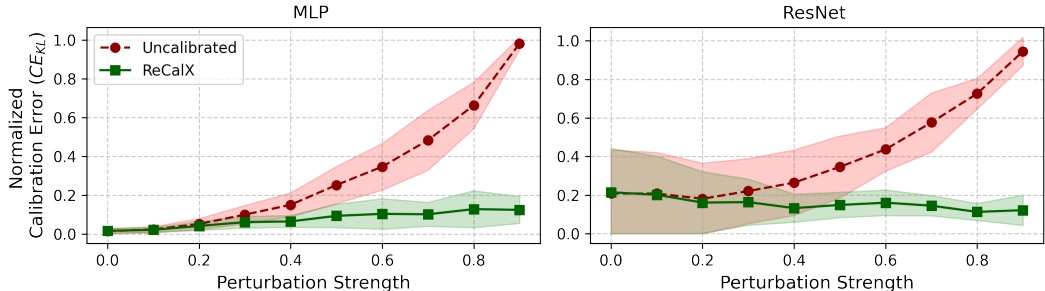

Figure 2: Normalized calibration errors aggregated across 10 tabular datasets for an MLP (left) and a ResNet model (right), including 95% confidence intervals. For both models, the miscalibration under the mean replacement perturbation tends to increase uniformly with higher perturbation levels.

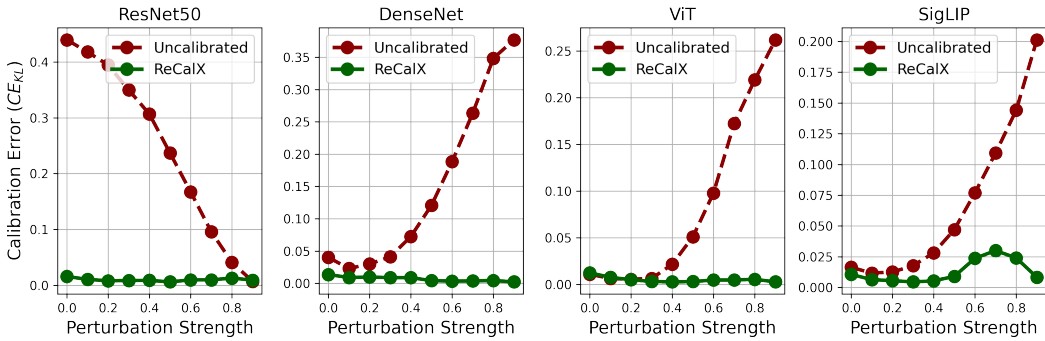

Figure 3: Calibration error results for popular image classifiers on ImageNet under fixed baseline perturbation with zeros. Across all methods, miscalibration varies significantly across the perturbation severity. While also for most image models the error tends to grow with perturbation level, for a ResNet50, miscalibration is worst for lower levels. This flexible behavior highlights the importance of calibration strategies that are adaptive to the perturbation strength.

**Global Remove and Retrain Fidelity** As a first experiment, we evaluated whether ReCalX improves the ability to identify truly important features that a model relies on. This global fidelity can be assessed through remove-and-retrain experiments [29, 12], where features are sequentially removed according to their importance ranking, and the model is retrained to measure the actual performance degradation caused. Higher performance drops indicate more accurate identification of genuinely important features. Figure 4 shows corresponding results across several tabular datasets for a Multi-Layer Perceptron (MLP) and a tabular ResNet architecture [22]. For both models, we used Shapley Values [42] with feature mean replacement as a perturbation and derived global importances as an average over 1000 samples. As suggested by [29], we performed retraining over 3 different random seeds and report the average loss increases across all runs per dataset. ReCalX consistently leads to importance rankings that, when used to drop features, cause a steeper loss surge compared to the original model's explanations. This aligns with our theoretical analysis in Section 3, which established

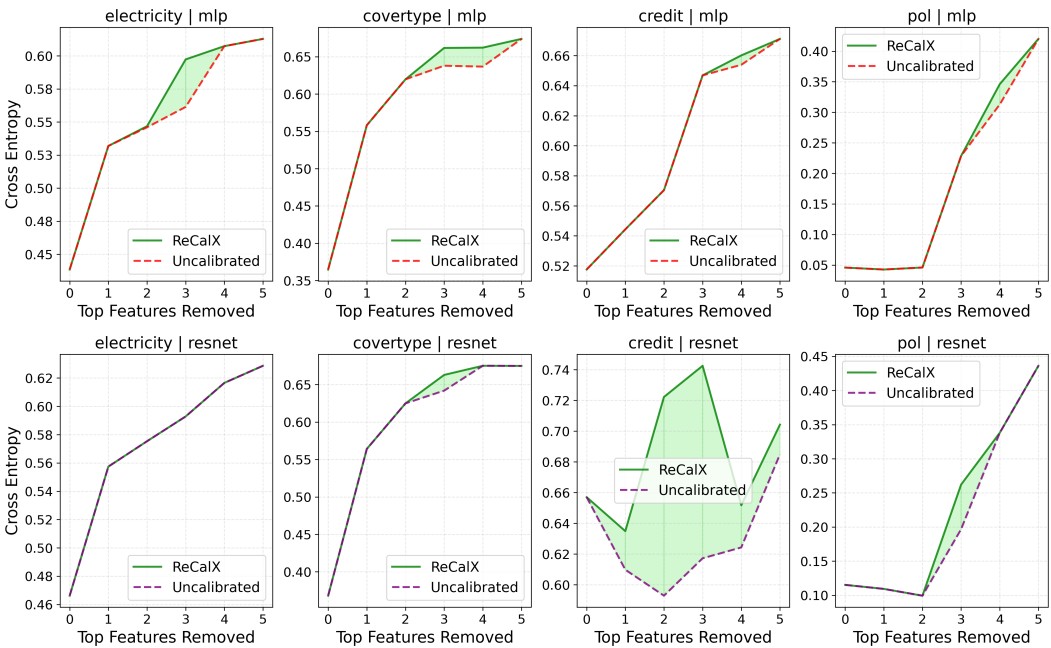

Figure 4: Retraining results on four tabular datasets for an MLP (top row) and a ResNet (bottom row) when the features are removed based on their global importance estimated via Shapley Values. Whenever calibrated explanations imply a different importance ranking (green area), the resulting performance loss is consistently higher compared to the uncalibrated importance indications. Hence, ReCalX enables better identification of truly relevant features that are crucial for good performance.

that calibration errors directly undermine the predictive power assessment of feature subsets. By reducing perturbation-specific calibration errors, ReCalX enables more accurate estimation of mutual information between feature subsets and the target variable, resulting in importance rankings that better reflect the true contribution of features to model performance. For instance, on the electricity dataset, removing the top three features ranked by ReCalX-enhanced explanations increased the original loss by 33%, compared to only 24% when using the original model's explanations. In Appendix C, we provide similar results on six additional datasets as well as for global importances obtained via LIME [53], all confirming the effectiveness of ReCalX in this regard.

**Explanation Robustness**     Perturbation-based techniques can lack stability when subjected to minor input variations, resulting in unreasonably different explanation results for seemingly equivalent predictions [1, 38, 3]. Sensitivity metrics [64, 5, 28] are a popular way to quantify explanation robustness by measuring how explanations change under tiny input modifications. In Table 3 we present Average and Max Sensitivity results across multiple image classifiers and explanation methods, each evaluated using 200 random samples from the ImageNet validation set. ReCalX substantially improves both metrics for all analyzed models, explanation methods, and both considered perturbation types. This indicates that proper recalibration also promotes explanation robustness, tracing one potential source of instabilities back to miscalibration effects.

**Visualizing the Effect of ReCalX on Explanations**     Finally, we show how ReCalX can impact explantion results on a qualitative level. In Figure 5 we display four representative examples where the calibrated explanation deviated substantially from the original one for an image classifier. All explanations have been computed based on Shapley Values with zero replacement on a DenseNet121 model (top row) and a SigLIP zero-shot model (bottom row) respectively. The examples demonstrate that after calibration with RecalX, the indicated feature importances are more concentrated around the actual object of interest, which typically contains the discriminative information with respect to its target label. Moreover, the calibrated ones also appear less noisy in general, which may contribute to higher explanation robustness in line with the experimental results reported above.

| Model | Zero Baseline Perturbation | | | | | | Blur Perturbation | | | | | |
|---|---|---|---|---|---|---|---|---|---|---|---|---|
| | LIME | | Kernel SHAP | | Feature Ablation | | LIME | | Kernel SHAP | | Feature Ablation | |
| | Uncal. | ReCalX | Uncal. | ReCalX | Uncal. | ReCalX | Uncal. | ReCalX | Uncal. | ReCalX | Uncal. | ReCalX |
| **Average Sensitivity ($S_{\text{AVG}} \downarrow$)** | | | | | | | | | | | | |
| ResNet50 | 1.349 | **1.190** | 1.434 | **1.364** | 0.965 | **0.825** | 1.321 | **1.202** | 1.403 | **1.335** | 0.965 | **0.845** |
| DenseNet121 | 1.174 | **0.952** | 1.465 | **1.125** | 0.716 | **0.602** | 1.224 | **1.118** | 1.500 | **1.425** | 0.602 | **0.556** |
| ViT | 1.498 | **1.155** | 1.399 | **1.279** | 1.041 | **0.880** | 1.399 | **1.224** | 1.631 | **1.571** | 0.905 | **0.785** |
| SigLIP | 1.215 | **0.963** | 1.434 | **1.222** | 1.140 | **0.922** | 1.518 | **1.349** | 1.714 | **1.659** | 1.140 | **1.015** |
| **Maximum Sensitivity ($S_{\text{MAX}} \downarrow$)** | | | | | | | | | | | | |
| ResNet50 | 1.914 | **1.595** | 2.873 | **2.143** | 1.247 | **1.025** | 2.010 | **1.658** | 2.507 | **2.035** | 1.247 | **1.061** |
| DenseNet121 | 1.649 | **1.101** | 2.649 | **1.535** | 0.777 | **0.776** | 1.760 | **1.525** | 2.779 | **2.404** | 0.777 | **0.701** |
| ViT | 2.339 | **1.571** | 2.309 | **2.028** | 1.467 | **1.126** | 2.076 | **1.746** | 2.998 | **2.796** | 1.246 | **1.032** |
| SigLIP | 1.737 | **1.093** | 2.729 | **1.787** | 1.606 | **1.033** | 2.327 | **1.921** | 3.493 | **2.897** | 1.606 | **1.379** |

Table 3: Average ($S_{\text{AVG}}$) and Maximum ($S_{\text{MAX}}$) sensitivity results for different perturbation-based explanation methods across diverse image classifiers on ImageNet. Results are shown for both zero baseline and blur perturbation techniques. Lower values indicate better robustness. ReCalX consistently improves robustness compared to uncalibrated (Uncal.) explanations in all cases.

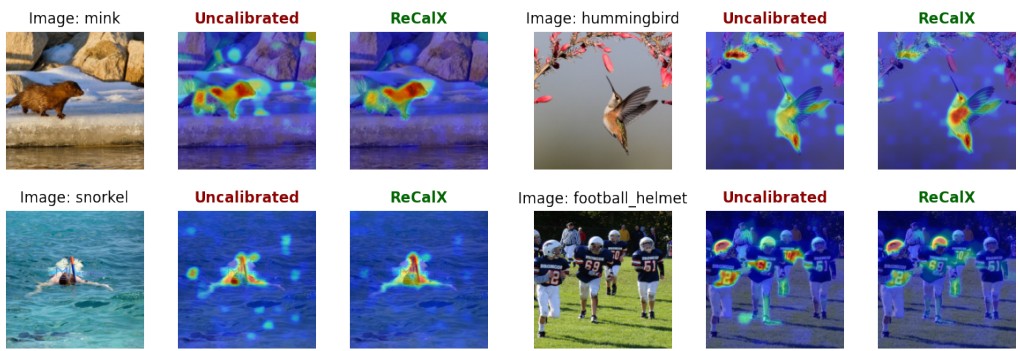

Figure 5: Qualitative comparison of Shapley Value explanations before and after ReCalX calibration for a DenseNet121 (top row) and a SigLIP zero-shot model (bottom row). The calibrated explanations exhibit stronger focus on discriminative object regions and reduced noise, demonstrating how miscalibration correction leads to more informative and robust feature attributions.

## 6 Discussion and Conclusion

In this work, we established a fundamental connection between uncertainty calibration and the reliability of perturbation-based explanations. Our theoretical analysis reveals that poor calibration under feature perturbations directly impacts explanation quality, leading to potentially misleading interpretations. To address this, we introduced ReCalX, a novel recalibration approach that specifically targets explanation-relevant miscalibration. Our empirical results demonstrate that ReCalX substantially reduces calibration error across different perturbation levels and consistently improves explanation quality, as measured by robustness and retraining fidelity. Although our approach is tailored particularly for perturbation-based methods, the underlying principles may also have implications for other explanation families, such as gradient-based [2] or counterfactual [62] methods, pointing towards potential directions for future work. While ReCalX requires additional computational efforts to be properly set up, including access to validation samples, it only introduces a minor extra load during inference arising from adaptive temperature selection when deriving explanations. However, this overhead is typically in the order of milliseconds (see Appendix D for corresponding experiments), and we believe that this is well justified to not only obtain more reliable predictions but also better explanations via uncertainty calibration.

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
