# OpenReview forum: "Improving Perturbation-based Explanations by Understanding the Role of Uncertainty Calibration"
_NeurIPS.cc/2025/Conference — NeurIPS 2025 spotlight_

### Official Review · Reviewer_cRdn · 2025-06-26

**Clarity:** 4
**Significance:** 3
**Originality:** 3
**Rating:** 6
**Confidence:** 4

**Summary:**

The paper analyzes how poor calibration affects the quality of perturbation-based explanations. To address this issue, and improve these explanation, ReCalX is introduced, a novel model calibration method tailored towards improving reliability of perturbation-based XAI, based on temperature scaling. Both the theoretical analysis and the efficacy (improved explanation robustness and fidelity) of ReCalX is validated through extensive experiments on a wide range of models and datasets.

**Questions:**

- How does the number of bins affect (1) the required number of validation samples and (2) calibration errors and resulting explanation quality.
- Could you include some qualitative examples of the effects of ReCalX (e.g., in the appendix)?
- Can the perturbation level $\lambda(S)$ also be computed for non-binary perturbations? If not, could it be extended to include these as well?
- Could you elaborate on the additional computational cost induced by ReCalX?

**Ethical Concerns:**

["NO or VERY MINOR ethics concerns only"]

**Final Justification:**

I am satisfied with the answers provided to mine and the other reviewers' comments and am therefore raising my score.

**Limitations:**

Yes

**Paper Formatting Concerns:**

No concerns

**Quality:**

4

**Strengths And Weaknesses:**

**Strengths.** The paper is extremely well written (including intuitive descriptions for equation-heavy theoretical results) and structured. The addressed issue is clearly and rigorously analyzed and explained. Experimental validation is extensive, relevant, self-contained, and convincing. The shown effects are (very) significant. Limitations are addressed. Especially the theoretical analysis of the effect of poor model calibration on (perturbation-based) explanations is of significant interest to the community, as it identifies a serious issue, with clear directions how to resolve it.

**Weaknesses.** Tbh, I struggle to find serious weaknesses with this work. The proposed solution, ReCalX is slightly incremental, as it simply optimizes the temperature parameter for bins of perturbation levels. The effect of hyperparameters of this approach is not analyzed further. E.g., how does the number of bins affect (1) the required number of validation samples and (2) calibration errors and resulting explanation quality.
Despite the quantitative analysis being convincing w.r.t. potential benefits of ReCalX, I would also have liked to have seen some qualitative example of how ReCalX affects the explanations. Beyond binary perturbations (such as setting features to zero or leaving them unperturbed), the exact computation of the perturbation level $\lambda(S)$ is not clear to me. Is there a way to consider also more continuous perturbations, such as e.g. perturbing features by inducing an alpha value? While the additional computational overhead induced by ReCalX is mentioned as a limitation, it is not quantified, which would aid in estimating the seriousness of this limitation.

---

> ### Author Rebuttal · Authors · 2025-07-31
>
> We highly appreciate the positive feedback and want to thank the reviewer for their insightful questions about ReCalX's implementation details and practical considerations.
>
> We've conducted additional analyses to address your concerns. Regarding bin count effects, our ablation study **shows increasing calibration improvements with higher bin counts, with diminishing returns**. The **number of bins doesn't directly affect validation sample requirements**, as we can generate sufficient perturbed samples for each bin. We agree that qualitative examples would enhance understanding of ReCalX's impact and **will include additional visualizations** in the revised manuscript. We've also clarified that our perturbation level concept **extends to non-binary perturbations** through a generalized intensity measure. Finally, by precisely quantifying inference times, we confirm that ReCalX adds **negligible computational overhead** during explanation generation in the order of milliseconds. All these findings will be included in the revised version of the manuscript.
>
> Below, we provide detailed responses to each of your questions.
>
> Q1: How does the number of bins affect (1) the required number of validation samples and (2) calibration errors and resulting explanation quality?
>
> A: Thank you for this relevant question, we will address both parts.
>
> The number of bins does not directly affect the required number of validation samples. Since one typically has sufficient control over the perturbation function when deriving explanations, we can generate a sufficient number of perturbed samples for any given perturbation level (bin). This ensures each bin has enough data for a stable temperature estimate, making the choice of bin count rather independent of the size of the available validation set.
>
> To investigate the effect on calibration errors, we conducted a dedicated ablation study. Using a Vision Transformer (ViT) model on ImageNet with the same experimental setup described in our main paper, we systematically varied the number of bins from 1 up to 25.
>
> | Number of Bins | ReCalX $CE_{KL}^{avg}$ | ReCalX $CE_{KL}^{max}$ |
> |----------------|------------------------|------------------------|
> | 1          	| 0.0506             	| 0.0915             	|
> | 3          	| 0.0137             	| 0.0440             	|
> | 5          	| 0.0092             	| 0.0215             	|
> | 10         	| 0.0075             	| 0.0138             	|
> | 15         	| 0.0070             	| 0.0128             	|
> | 20         	| 0.0070             	| 0.0116             	|
> | 25         	| 0.0067             	| 0.0116             	|
>
> The results reveal that increasing the number of bins generally improves calibration performance, particularly by reducing the maximum calibration error ($CE_{KL}^{max}$). This is expected because a finer-grained binning strategy allows ReCalX to learn a more nuanced set of temperatures that better match the model's varying calibration properties across different perturbation levels.
>
> However, we observe diminishing returns, with the most significant improvements occurring up to about 10 bins. Beyond this point, the reduction in calibration error becomes increasingly marginal, and therefore, 10 offers a balanced choice. As established by our theoretical results, this direct reduction in calibration error translates to more robust and faithful explanations.
>
> Q2: Could you include some qualitative examples of the effects of ReCalX (e.g., in the appendix)?
>
> A: Thank you very much for this great suggestion. We completely agree that qualitative examples are helpful for illustrating the practical impact of our method. One qualitative example was already provided as part of Figure 1 but we are happy to include additional ones in the revised version of the manuscript.
>
> Q3: Can the perturbation level also be computed for non-binary perturbations? If not, could it be extended to include these as well?
>
> A: Thank you for this excellent question. Yes, the concept of a perturbation level is highly flexible and can be generalized beyond binary perturbations.
>
> The fraction of perturbed features we use is a specific instance of a more general principle. For any perturbation strategy, we can define a per-feature perturbation intensity $\alpha_i \in [0, 1]$, where $\alpha_i=0$ means the feature is unchanged and $\alpha_i=1$ means it is maximally perturbed.
>
> The overall perturbation level $\lambda$ can then be universally defined as the average intensity across all $d$ features:
>
> $$
> \lambda = \frac{1}{d}\sum_i \alpha_i
> $$
>
> This formulation naturally extends binary perturbations where $\alpha_i \in$ {0, 1} to continuous ones (e.g., varying blur strength or noise intensities), ensuring ReCalX is broadly applicable.
>
> Q4: Could you elaborate on the additional computational cost induced by ReCalX?
>
> A: Thank you for this important practical question.
>
> The additional computational cost of ReCalX at inference time —when an explanation is actually generated— is negligible.
> This is because the total explanation time is overwhelmingly dominated by the repeated model queries required by the perturbation-based method itself. In contrast, ReCalX only adds two trivial operations: a quick temperature lookup and a single division on the model's logits.
>
> To quantify this, we measured the per-explanation overhead for various models on ImageNet. The table below shows the additional computation time in seconds induced by ReCalX when computing different explantion metods as average over 100 runs. The results confirm that the extra time is minor.
>
> | Method        	| ResNet18  | DenseNet  | ViT  	| SigLip |
> |------------------|-----------|-----------|----------|------------|
> | LIME        	| 0.0193s   | 0.0219s   | 0.0184s  | 0.0210s	|
> | Shapley Values| 0.2587s   | 0.2517s   | 0.1754s  | 0.3431s    |

---

> > ### Comment · Reviewer_cRdn · 2025-08-04
> >
> > Thank you for the detailed rebuttal. I am satisfied with the answers and have therefore reconsidered my score.

---

### Official Review · Reviewer_v21P · 2025-06-30

**Clarity:** 3
**Significance:** 2
**Originality:** 3
**Rating:** 4
**Confidence:** 3

**Summary:**

This paper explores how uncertainty miscalibration under input perturbations can degrade the reliability of perturbation-based explanations. The authors introduce **ReCalX**, an adaptive temperature scaling method that recalibrates model confidence specifically for perturbed inputs without altering predictions. Theoretical analysis shows that miscalibration directly impacts explanation quality, and ReCalX is designed to preserve information while reducing perturbation-specific calibration error. Empirical results across tabular and image datasets demonstrate that ReCalX improves explanation fidelity and robustness compared to standard calibration methods.

**Questions:**

Question: How does ReCalX perform under perturbation strategies that are not based on mean replacement or simple blurring (e.g., adversarial, manifold-preserving, or semantically grounded perturbations)?
Suggestion: Please discuss or evaluate ReCalX in such settings to assess generality. Including even a small experiment could strengthen the claims of robustness.

Question: How sensitive is ReCalX to the size and representativeness of the validation set used for estimating perturbation-level calibration errors?
Suggestion: Clarify whether performance degrades with smaller or distribution-shifted calibration sets, and if any data-efficient variants could be used.

Question: How compatible is ReCalX with other classes of explanation methods (e.g., gradient-based, counterfactual explanations)?
Suggestion: While the focus is on perturbation-based methods, a brief discussion of limitations or extensions to other explanation families would be helpful.

**Ethical Concerns:**

["NO or VERY MINOR ethics concerns only"]

**Limitations:**

The authors briefly mention limitations in the conclusion, but a more explicit discussion would be valuable.

- The need for perturbation-specific validation data may limit ReCalX’s applicability in settings where such data is hard to obtain or generate.
- The computational overhead of applying different temperature scalings at inference time could be a concern for large-scale or real-time deployments.
- The method is designed specifically for perturbation-based explanations, and it is unclear how it generalizes to other explanation paradigms (e.g., gradient-based, counterfactual).

**Paper Formatting Concerns:**

No major formatting issues were found.

**Quality:**

3

**Strengths And Weaknesses:**

Strengths:

1. Strong theoretical foundation and thorough experiments across diverse models and datasets.
2. Well-written and easy to follow, with helpful figures and explanations.
3. Addresses an important and underexplored issue—miscalibration in perturbation-based explanations.
4. First method to explicitly recalibrate for explainability; novel connection between calibration and explanation quality.

Weaknesses:

1. Limited evaluation on complex or domain-specific perturbations.
2. Assumes access to validation data under perturbations, which may not always be feasible.
3. Slight increase in computational cost during explanation due to adaptive scaling.

---

> ### Author Rebuttal · Authors · 2025-07-31
>
> We want to thank the reviewer for their thoughtful questions and constructive comments.
>
> Based on your suggestions, we have conducted **additional experiments** to address your concerns about the **generality and robustness of ReCalX**. For more complex perturbation strategies, we evaluated ReCalX with **advanced inpainting techniques** and found it **remains highly effective in reducing calibration errors**. Regarding validation set sensitivity, our ablation study shows that even **small sets of 50-100 samples achieve most of the potential calibration improvement**, making ReCalX **practical in resource-constrained settings**. We've also **clarified the relationship between ReCalX and other explanation families** like gradient-based and counterfactual methods, identifying potential extensions for future work. Finally, we've quantified the computational overhead of ReCalX, **confirming it adds minimal processing time** during explanation generation. These findings strengthen our claims about ReCalX's effectiveness and practicality across diverse settings and will be included in the revised version of the paper.
>
> Detailed responses to each point follow below.
>
> Q1: How does ReCalX perform under perturbation strategies that are not based on mean replacement or simple blurring (e.g., adversarial, manifold-preserving, or semantically grounded perturbations)? Suggestion: Please discuss or evaluate ReCalX in such settings to assess generality. Including even a small experiment could strengthen the claims of robustness.
>
> A: Thank you for this insightful question and the concrete suggestion to approach it..
> We followed your advice and conducted an extra experiment evaluating ReCalX with more complex, domain-aware perturbation strategies. We chose two advanced image inpainting techniques available in the official shap library, which leverage the cv2.inpaint_telea and cv2.inpaint_ns algorithms provided by the cv2 package. Unlike simple blurring or zero-filling, these methods are designed to fill corrupted image regions in a semantically plausible way by incorporating information from the surrounding pixels.
>
> We evaluated the performance of ReCalX on a Vision Transformer (ViT) model when using these inpainting methods. The table below shows the average and  maximum KL-divergence calibration errors before and after applying ReCalX as well as for basic temperature scaling.
>
> | Perturbation   | Initial_Avg | Initial_Max | TS_Avg | TS_Max | ReCalX_Avg | ReCalX_Max |
> |----------------|-------------|-------------|--------|--------|------------|------------|
> | inpaint_ns 	| 0.3796  	| 1.0337  	| 0.2327 | 0.6890 | **0.0305**	| **0.1012** 	|
> | inpaint_telea  | 0.3843  	| 1.0612  	| 0.2370 | 0.6993 | **0.0293** 	| **0.0991** 	|
>
> The results demonstrate that ReCalX remains highly effective and substantially reduces the calibration error even for these sophisticated, domain-specific perturbations.
>
> Q2: How sensitive is ReCalX to the size and representativeness of the validation set used for estimating perturbation-level calibration errors? Suggestion: Clarify whether performance degrades with smaller or distribution-shifted calibration sets, and if any data-efficient variants could be used.
>
> A: Thank you for this important question. We investigated the sensitivity of ReCalX to the size and representativeness of the validation set.
>
> We performed an ablation study on a ViT model on ImageNet, varying the number of samples in the calibration set. The uncalibrated baseline errors were 0.0936 (Average) and 0.2618 (Max) and below we show the average results across 10 random seeds:
>
> | Cal Size | ReCalX $CE_{KL}^{avg}$ | ReCalX $CE_{KL}^{max}$ |
> |----------|----------------|----------------|
> | 10   	| 0.0175     	| 0.0743     	|
> | 25   	| 0.0099     	| 0.0346     	|
> | 50   	| 0.0045     	| 0.0115     	|
> | 75   	| 0.0035     	| 0.0088     	|
> | 100  	| 0.0031     	| 0.0063     	|
> | 200  	| 0.0028     	| 0.0060     	|
> | 500  	| 0.0021     	| 0.0027     	|
> | 1000 	| 0.0021     	| 0.0025     	|
>
> As the table shows, even a small, randomly selected validation set of 50-100 samples is sufficient to achieve the vast majority of the potential calibration improvement. Because measurable gains are already achievable with a small random set, we conclude that specialized data-efficient selection techniques are not necessary for the practical application of our method.
>
> Regarding representativeness, ReCalX follows standard machine learning practice: it performs optimally when the validation set is drawn from the same underlying distribution as the data for which explanations will be generated. However, a key advantage of our setting is that we have full control over the perturbation function. This allows us to create a calibration set that is perfectly representative of the perturbed data distribution the model will face during the explanation process, mitigating concerns about unexpected distribution shifts.
>
> Q3: How compatible is ReCalX with other classes of explanation methods (e.g., gradient-based, counterfactual explanations)? Suggestion: While the focus is on perturbation-based methods, a brief discussion of limitations or extensions to other explanation families would be helpful.
>
> A: Thank you for this insightful question, which helps clarify the scope and potential future directions of our work.
> While our method is tailored particularly tailored for perturbation-based methods, the underlying principle has indeed connections to other explanation families.
>
> For instance, it may also be relevant for gradient-based methods like SmoothGrad and Integrated Gradient, which compute explanations by averaging gradients over perturbed inputs. Therefore, ensuring the model is well-calibrated on these very perturbations could also lead to more stable and reliable gradient signals, directly improving the final explanation. Exploring how calibration properties affect such kinds of explanations is an interesting direction for future work.
>
> Moreover, counterfactual explanation methods search for a new data point $x'$ in the proximity of a point $x$ that yields a desired prediction. Evaluating the reliability of the model's output on the generated counterfactual $x'$ could also have an influence on explanation quality. However, ensuring the model is well-calibrated on $x'$ addresses a different problem than ReCalX, which corrects for a given perturbation function used to derive explanation. But analyzing calibration in the context of counterfactuals may also be an interesting direction for future work.
> We will include a corresponding discussion on the relationship to other explanation families in the revised version of the manuscript.
>
> Next we further provide answers to all additional weakness identified in the review:
>
> W2: Assumes access to validation data under perturbations, which may not always be feasible.
>
> A: Thank you for mentioning this limitation, which we acknowledged in our conclusion. However, relying on at least a limited amount of validation data is a very common assumption that underlies many machine learning approaches. Moreover, we now have also performed an ablation study (see Q3 above)  indicating that even with a small number of validation samples, ReCalX can yield significant gains in calibration under the perturbation used during the explanation process. Hence, we argue that the severity of this limitation is minor.
>
> W3: Slight increase in computational cost during explanation due to adaptive scaling.
>
> A: Thank you also for bringing this up. To quantify this, we measured the per-explanation overhead for various models on ImageNet. The table below shows the additional computation time in seconds induced by ReCalX when computing different explantion metods as average over 100 runs. The results confirm that the extra time is indeed minor. For more details, please also refer to the answer of Q3 provided to reviewer cRdn.
>
> | Method        	| ResNet18  | DenseNet  | ViT  	| SigLip |
> |------------------|-----------|-----------|----------|------------|
> | LIME        	| 0.0193s   | 0.0219s   | 0.0184s  | 0.0210s	|
> | Shapley Values| 0.2587s   | 0.2517s   | 0.1754s  | 0.3431s    |

---

### Official Review · Reviewer_idkw · 2025-07-02

**Clarity:** 4
**Significance:** 3
**Originality:** 3
**Rating:** 5
**Confidence:** 3

**Summary:**

This paper makes a key contribution by establishing the first comprehensive relationship between uncertainty calibration and the reliability of perturbation-based explanations. The authors introduce ReCalX, a novel and practical solution designed to enhance the quality of these explanations. Grounded in the theoretical principle of preserving information while reducing calibration error under perturbation, ReCalX employs a unique temperature scaling method sensitive to both perturbation level and intensity. Extensive experiments are conducted to validate the proposed approach.

**Questions:**

The theory presented is confined to linear summary strategies and classification problems. It remains an open question how these findings generalize to non-linear aggregation methods (e.g. use decision trees as surrogates or others, whether the boundedness property and Operatornorm are still valid) and how ReCalX would perform in that context. I am willing to reconsider my score if the authors provide more intuition or justification for the extensions.

**Ethical Concerns:**

["NO or VERY MINOR ethics concerns only"]

**Final Justification:**

The rebuttal effectively addresses my questions. It strengthens the paper's generalizability by extending the theory to include non-linear aggregation strategies and general prediction tasks. Therefore, I've decided to raise my score.

**Limitations:**

(minor) can discuss the potential extension to regression problems and explanations with non-linear summaries.

**Quality:**

4

**Strengths And Weaknesses:**

The paper is well-written and the study is well-executed. The motivation, theory, and method are all stated clearly, and the experimental results align well with the theoretical claims. The work is also novel, as it is the first to provide a theoretical basis for the relationship between uncertainty calibration and perturbation-based explanations. Following this, the proposed ReCalX method is a novel and promising approach that incorporates perturbation level and intensity for calibration and reliable explanations.

---

> ### Author Rebuttal · Authors · 2025-07-31
>
> We want to thank the reviewer for their thoughtful comments and suggestions.
>
> We have addressed both key concerns raised in your review. Regarding non-linear aggregation methods, we demonstrate that our **theoretical framework fully extends beyond linear aggregation strategies** by reformulating our bounds using a general Lipschitz property. We also **explain how our theory naturally applies to general prediction tasks** beyond classification. For regression problems, we've conducted **additional experiments on tabular regression datasets** that show ReCalX **effectively reduces miscalibration under explanation-specific perturbations in regression tasks** as well. These extensions and experimental results will be incorporated into the revised manuscript to provide a more comprehensive treatment of our approach.
>
> Detailed responses to each point are provided below.
>
> W1: It remains an open question how these findings generalize to non-linear aggregation methods
>
> A: Thank you for this important remark, as it helps clarify a central aspect of our work.
> In fact, all the theoretical results and conclusions in our paper can be extended to also hold for perturbation-based methods with non-linear aggregation strategies.
>
> For our first theoretical result (Theorem 3.2 and Corollary 3.3), which decomposes predictive power into information-theoretic components, the analysis is completely independent of the aggregation strategy. This theorem applies universally to any perturbation-based approach since it focuses on the fundamental relationship between calibration error and predictive power.
>
> Regarding Theorem 3.4 (local explanation bounds), while we initially presented it for linear aggregation strategies due to their popularity, the result can be naturally extended to non-linear aggregation methods. For any general non-linear aggregation function $G$ that maps model predictions under perturbations to feature importance vectors, we can reformulate the bound using a more general Lipschitz property.
>
> Let $G$ be a non-linear aggregation strategy, and let $\nu(x)$ and $\nu^*(x)$ represent the vectors of model predictions under subset perturbations for the uncalibrated and perfectly calibrated models, respectively. All we additionally need for the non-linear case is the existence of a constant L such that:
>
> $$
> \lVert G(\nu(x)) - G(\nu^*(x)) \rVert_2 \le L \lVert \nu(x) - \nu^{\ast}(x) \rVert_{\infty}
> $$
>
> This leads to the following more general version of Theorem 3.4:
>
> $$
> \frac{1}{d}\lVert \phi(x) - \phi^*(x) \rVert_2^2 \le \dfrac{L^2}{d} CE_{\textit{KL}}^{\max_S} + \sqrt{\frac{L^4}{2d}\log(1/\delta)}
> $$
>
> For linear aggregation strategies used by methods like SHAP and LIME, one can show that $L = 2\sqrt{d}$ [1], which yields the specific bound presented in Theorem 3.4 of the paper.
>
> This demonstrates that our theoretical framework generalizes to non-linear aggregation strategies as well. We will add a corresponding discussion to the revised version of the paper.
>
> [1] Lin, Chris, Ian Covert, and Su-In Lee. "On the robustness of removal-based feature attributions." Advances in Neural Information Processing Systems 36 (2023): 79613-79666.
>
> W2: can discuss the potential extension to regression problems
>
> A: Thank you for this thoughtful suggestion. We primarily focus on classification problems in the main paper due to their popularity in the explainability and calibration literature. However, our theoretical framework and the ReCalX approach of performing perturbation-specific recalibration can be extended to regression problems as well.
>
> First notice, that the core principles of our paper are information-theoretic and generalize to any predictive modeling task. For regression, we consider a model $f: \mathcal{X} \to \mathcal{P}(\mathcal{Y})$ that maps to a continuous target space $\mathcal{Y} \subseteq \mathbb{R}$ with continuous distribution $P_Y$. In this setup, all theoretical statements developed in section 3 of the paper hold equivalently when explaining the propability of an output with perturbation-based methods.
>
> Moreover, also the notion of information preserving calibration and the concept of strictly monotonic transformations to preserve the model’s learned information content extend naturally.  While temperature scaling is indeed tailored to classification settings, techniques like affine transformations [3] or injective versions of Isotonic Regression [4] satisfy this property more generally and are also applicable to regression problems.
>
> To demonstrate this compatibility, we conducted experiments on two tabular regression datasets to evaluate miscalibration under fixed baseline replacement with the feature mean. Following a setup equivalent to our main experiments, we trained a simple MLP regression model with gaussian output distribution [5] and applied ReCalX by learning a simple Platt-scaling like affine transformation (a scaling and a shift parameter) for each perturbation level.
> The table below show average and maximum quantile-based calibration errors [2] across perturbation strengths.
>
> | Dataset  | Uncalibrated $CE^{AVG_S}$  | ReCalX $CE^{AVG_S}$ | Uncalibrated $CE^{MAX_S}$  | ReCalX  $CE^{MAX_S}$  |
> |--------------|----------------------------|---------------------------|----------------------------|--------------------------|
> | wine_quality   | 0.06495                       	| **0.01182**                       	| 0.1053                          	|**0.0183**                 |
> | houses         	| 0.18078                       	|**0.00741**                       	| 0.2512                          	|**0.0178**                 |
>
> The results indicate that ReCalX is also capable of reducing miscalibration under explainably-specific perturbations for regression tasks. We will provide a comprehensive discussion in the revised version of the manuscript.
>
> [1] Song, H., Diethe, T., Kull, M., and Flach, P. Distribution calibration for regression. In International Conference on Machine Learning, pp. 5897–5906. PMLR, 2019.
>
> [2]Kuleshov, Volodymyr, Nathan Fenner, and Stefano Ermon. "Accurate uncertainties for deep learning using calibrated regression." International conference on machine learning. PMLR, 2018.
>
> [3] Platt, J. C. Probabilistic outputs for support vector machines and comparisons to regularized likelihood methods. In Advances in Large-Margin Classifiers, pp. 61–74. MIT Press, 1999.
>
> [4] Zhang, J., Kailkhura, B., and Han, T. Y.-J. Mix-n-match: Ensemble and compositional methods for uncertainty calibration in deep learning. In International Conference on Machine Learning, pp. 11117– 11128. PMLR, 2020.
>
> [5] Bishop, C. M. Mixture density networks. 1994

---

> > ### Comment · Reviewer_idkw · 2025-08-08
> > **Official Comment by Reviewer idkw**
> >
> > I thank the authors for their detailed reply. The provided theoretical framework extends well beyond linear aggregation strategies and clearly explains how the theory naturally applies to general prediction tasks. After reading the reply, I believe these discussions have made the paper even more comprehensive. Thus, I have raised my score.

---

### Official Review · Reviewer_CfX7 · 2025-07-03

**Clarity:** 4
**Significance:** 2
**Originality:** 3
**Rating:** 5
**Confidence:** 2

**Summary:**

This paper proposes a connection between uncertainty calibration and explanation performance for perturbation based explainability in linear subset aggregation methods. The authors show that calibration error can be directly connected to the predictive power for perturbation-based predictions, and can thus derail explainability approaches within this framework when not considered. A calibration framework based on temperature scaling is proposed to mitigate this.

**Questions:**

1. Is there anything about the theoretical results which requires linearity of the subset-based perturbation aggregations?
2. How sensitive is your recalX approach to the number of bins selected? Are there any tradeoffs to be aware of?
3. It seems that, given that the temperature scaling is tuned at validation time and that this tuning becomes more complex as the number of perturbations and binning levels increases, this approach may suffer from overfitting issues. Have you noticed or investigated robustness to overfitting of the tuned temperature scaling with respect to calibration error when the perturbations may change at test time?

**Ethical Concerns:**

["NO or VERY MINOR ethics concerns only"]

**Final Justification:**

The paper introduces a novel connection between uncertainty calibration and performance for perturbation based explainability approaches. While I initially recommended borderline acceptance, with some concerns remaining for the full generality of the results to non-linear aggregation methods and regression tasks, the authors have adequately addressed these concerns in their response. In particular, the authors have clarified that their theoretical contributions can be generalized to non-linear aggregation methods, in additional to general target distributions.

Assuming the author's clarifications and extended results are incorporated into the final manuscript, I am thus comfortable in more strongly recommending acceptance.

**Limitations:**

yes

**Paper Formatting Concerns:**

No formatting concerns.

**Quality:**

3

**Strengths And Weaknesses:**

## Strengths

1. The connection between calibration error and perturbation-based predictive power is interesting, and clearly has implications for how to design further explainability approaches.
2. I found the paper to be very clearly written with a good flow, particularly for the theoretical contents of the paper.
3. The authors discuss the implications of the work for both global and local explanations. Unfortunately the local explanation results may not have a very tight bound, however this is presumably to be expected and it is good for it to be discussed, as often in practice one will be interested in local explanations.

## Weaknessess
1. The proposed solution to the calibration problem, namely a binned temperature scaling, feels somewhat simplistic and is specific to classification tasks. It would have been nice to see more discussion about alternative ideas to address the calibration issues, for example exploring training-time approaches, particularly given that the temperature based recalibration approach will always be restricted to the classification setting.
2. Given that notions of calibration can be extended to regression, it would have been interesting to see a more complete discussion about the implications of the work in this setting also. Indeed, many relevant prediction tasks may also be for regression.
3. The paper focuses the experiments exclusively on linear subset aggregation explanation methods (LIME and Shapley). Is this required for the theory to hold? If not, it is odd to focus only on linear explanation methods when the context of the experiments focuses squarely on non-linear networks where complex inter-dependencies between features are likely to exist.

---

> ### Author Rebuttal · Authors · 2025-07-31
>
> We want to thank the reviewer for their thoughtful questions and helpful suggestions.
>
> Based on your feedback, we have now performed **additional ablation studies on the number of bins** for ReCalX and analyzed the **generalization capabilities across different perturbation types**. The results show **monotonic but saturating improvements in calibration with higher bin count** and demonstrates **strong correlation between calibration patterns across perturbation strategies**. We have also clarified that our **theoretical results extend to non-linear aggregation strategies**, and we address your concerns about the simplicity of our approach and its **applicability beyond classification settings**. These findings and clarifications will be included in the revised version of the paper.
>
> Detailed answers to all your questions and concerns are provided below.
>
> Q1: Is there anything about the theoretical results which requires linearity of the subset-based perturbation aggregations?
>
> A: Thank you for this important question, as it helps clarify a central aspect of our work.
> In fact, all the theoretical results and conclusions in our paper can be extended to also hold for perturbation-based methods with non-linear aggregation strategies.
> For our first theoretical result (Theorem 3.2 and Corollary 3.3), which decomposes predictive power into information-theoretic components, the analysis is completely independent of the aggregation strategy. This theorem applies universally to any perturbation-based approach since it focuses on the fundamental relationship between calibration error and predictive power.
> Regarding Theorem 3.4 (local explanation bounds), while we initially presented it for linear aggregation strategies due to their popularity, the result can be naturally extended to non-linear aggregation methods. For any general non-linear aggregation function $G$ that maps model predictions under perturbations to feature importance vectors, we can reformulate the bound using a more general Lipschitz property.
> Let $G$ be a non-linear aggregation strategy, and let $\nu(x)$ and $\nu^*(x)$ represent the vectors of model predictions under subset perturbations for the uncalibrated and perfectly calibrated models, respectively. All we additionally need for the non-linear case is the existence of a constant L such that:
>
> $$
> \lVert G(\nu(x)) - G(\nu^*(x)) \rVert_2 \le L \lVert \nu(x) - \nu^{\ast}(x) \rVert_{\infty}
> $$
>
> This leads to the following more general version of Theorem 3.4:
>
> $$
> \frac{1}{d}\lVert \phi(x) - \phi^*(x) \rVert_2^2 \le \frac{L^2}{d} CE_{\textit{KL}}^{\max_S} + \sqrt{\frac{L^4}{2d}\log(1/\delta)}
> $$
>
> For linear aggregation strategies used by methods like SHAP and LIME, one can show that $L = 2\sqrt{d}$ [1], which yields the specific bound presented in Theorem 3.4 of the paper.
> This demonstrates that our theoretical framework can be generalized to non-linear aggregation strategies as well. We will add a corresponding discussion to the revised version of the paper.
>
> [1] Lin, Chris, Ian Covert, and Su-In Lee. "On the robustness of removal-based feature attributions." Advances in Neural Information Processing Systems 36 (2023): 79613-79666.
>
> Q2: How sensitive is your recalX approach to the number of bins selected? Are there any tradeoffs to be aware of?
>
> A: Thank you for this insightful question regarding the sensitivity of ReCalX to the number of bins.
> We have conducted a dedicated ablation study to investigate this. Using a Vision Transformer (ViT) model on ImageNet with the same experimental setup described in our main paper, we systematically varied the number of bins from 1 up to 25.
>
> | Number of Bins | ReCalX $CE_{KL}^{avg}$ | ReCalX $CE_{KL}^{max}$ |
> |----------------|------------------------|------------------------|
> | 1          	| 0.0506             	| 0.0915             	|
> | 3          	| 0.0137             	| 0.0440             	|
> | 5          	| 0.0092             	| 0.0215             	|
> | 10         	| 0.0075             	| 0.0138             	|
> | 15         	| 0.0070             	| 0.0128             	|
> | 20         	| 0.0070             	| 0.0116             	|
> | 25         	| 0.0067             	| 0.0116             	|
>
> The results reveal that increasing the number of bins generally improves calibration performance. This is expected because a finer-grained binning strategy allows ReCalX to learn a more nuanced set of temperatures that better match the model's varying calibration properties across different perturbation levels.
> However, we observe diminishing returns, with the most significant improvements occurring up to about 10 bins. Beyond this point, the reduction in calibration error saturates.
>
> This presents a minor tradeoff between calibration granularity and computational cost during initial setup.
> A higher number of bins allows ReCalX to learn a more fine-grained set of temperatures, better adapting to the model's specific miscalibration at different perturbation levels. As shown in our ablation study, this generally leads to a lower maximum calibration error.
> However, each bin requires a separate temperature parameter to be optimized on the validation set. Therefore, increasing the number of bins linearly increases the computational efforts associated with the one-time calibration procedure.
> Our empirical results indicate that the performance gains from adding more bins diminish and eventually saturate. To manage this trade-off one can choose a number of bins (e.g., 10) that captures most of the potential calibration improvement without incurring unnecessary high computational cost during setup.
>
> Q3: Have you noticed or investigated robustness to overfitting of the tuned temperature scaling with respect to calibration error when the perturbations may change at test time?
>
> A: Thank you also for this insightful question.
>
> The risk of overfitting with ReCalX is inherently low due to the simplicity of our approach. We are only optimizing a very small number of parameters (e.g., 10 temperatures for 10 bins), which is a model of extremely low complexity. This makes it highly unlikely for the learned temperatures to overfit to statistical noise in the calibration data.
> Moreover, applying a perturbation-based explanation method typically assumes a predefined single, fixed perturbation strategy making changing perturbations at test time unusual. However, we also investigated the interesting question of whether the calibration learned for one perturbation type could generalize to another.
>
> During the experiments in the paper, we already optimized two separate sets of ReCalX temperatures for different classifiers on ImageNet: one using blur perturbations and another using zero-baseline replacement perturbations. This allows us to compute the correlation between the resulting sets of temperatures across the perturbation levels to assess their correspondance.
>
> | Model   	| Pearson Correlation 	|
> |-------------|-----------|
> | ResNet50	| 0.984  |
> | DenseNet	| 0.975  |
> | ViT	| 0.914  |
> | SigLip  | 0.956    |
>
> The results show a strong positive correlation, suggesting that a model's underlying miscalibration patterns can be similar across different perturbation types. While this indicates that ReCalX may offer some generalization, we recommend explicitly calibrating for the specific perturbation strategy being used. This ensures that the theoretical benefits of our method are fully realized for the explanation method at hand.
>
> Next,  we also provide individual responses to all additional weaknesses (W) mentioned in the review.
>
> W1.1: The proposed solution to the calibration problem, namely a binned temperature scaling, feels somewhat simplistic and is specific to classification tasks.
>
> A: Thank you for raising this point. We view the simplicity of our approach as a key strength, as it leads to a practical and efficient method that is easy to apply and due to its small number of parameters not prone to overfitting. Our experiments consistently show that ReCalX based on temperature scaling is highly effective, delivering substantial improvements in calibration and explanation quality with minimal overhead.
>
> W1.2: It would have been nice to see more discussion about alternative ideas to address the calibration issues, for example exploring training-time approaches, particularly given that the temperature based recalibration approach will always be restricted to the classification setting.
>
> A: Thank you for this suggestion. When applying perturbation-based explanations the goal typically is to explain an existing, already-trained model. Therefore, any calibration method we use should be information-preserving (Definition 4.1) to avoid altering the model's core logic. Training-time approaches would create a new, different model, which is not what we aim to explain. Importantly, ReCalX and its core idea to apply an adaptive, information-preserving calibration map during the explanation process generalizes beyond classification. We detail in our response to reviewer idkw below how for instance, affine transformations can be used for learning suitable calibration maps in the regression setting and provide experiments on 2 tabular regression datasets.
>
> W2: Given that notions of calibration can be extended to regression, it would have been interesting to see a more complete discussion about the implications of the work in this setting also.
>
> A: Thank you for mentioning this, we agree that discussing how ReCalX can be applied to regression problems would be helpful. In fact, all theoretical results and conclusions can be extended to arbitrary target distributions. We will include a comprehensive discussion on this in the revised version of the manuscript. For more details on this, we would like to refer to the answer to W 2 provided for reviewer idkw below, where we also provide additional experiments on tabular regression datasets that show ReCalX efficacy in regression tasks as well.

---

> > ### Comment · Reviewer_CfX7 · 2025-08-01
> >
> > Thank you for your detailed response, and in particular for the clarifications regarding the generalization in theorem 3.4 to non-linear aggregation approaches. Given that my concerns have been adequately addressed, I am comfortable in increasing my score, under the assumption that the authors will include their more general results, and discussion for more general target distributions, to the final manuscript.

---

### Decision · Program_Chairs · 2025-09-17

**Decision:**

Accept (spotlight)

**Comment:**

This paper establishes a connection between uncertainty calibration and explanation quality in perturbation-based explainability for linear aggregation methods.
The authors demonstrate that calibration error can undermine explanation reliability and propose a temperature scaling framework to address this.
They introduce ReCalX, a calibration method designed to enhance the robustness and fidelity of perturbation-based explanations.
Theoretical analysis and comprehensive experiments support the effectiveness of ReCalX.
While the theory focuses on linear aggregation methods, the authors clarified that it could be extended beyond the linear case.